# Mobilization and recycling of intracellular phosphorus in response to availability

Chih-Pin Chiang[1], Joseph Yayen[1] and Tzyy-Jen Chiou[1]

[1] Agricultural Biotechnology Research Center, Academia Sinica, Taipei, Taiwan

phosphate transporter; phosphorus recycling; phosphorus remobilization.

**Corresponding author**:
Tzyy-Jen Chiou.
Email: tjchiou@gate.sinica.edu.tw

**Associate Editor:** Ingo Dreyer

## Abstract

Phosphorus (P) is a non-renewable resource that limits plant productivity due to its low bioavailability in the soil. Large amounts of P fertilizer are required to sustain high yields, which is both inefficient and hazardous to the environment. Plants have evolved various adaptive responses to cope with low external P availability, including mobilizing cellular P through phosphate ($P_i$) transporters and recycling $P_i$ from P-containing biomolecules to maintain cellular P homeostasis. This mini-review summarizes the current research on intracellular P recycling and mobilization in response to P availability. We introduce the roles of $P_i$ transporters and the P metabolic enzymes and expand on their gene regulation and mechanisms. The relevance of these processes in the search for targets to improve phosphorus use efficiency and some of the current challenges and gaps in our understanding of P starvation responses are discussed.

## 1. Introduction

Phosphorus (P) is a constituent of essential biomolecules for plant growth and survival (Lambers, 2022). Inorganic orthophosphate ($H_2PO_4^-$, $HPO_4^{2-}$; $P_i$) is the predominant form of P directly acquired by plant roots. However, $P_i$ is limited by its low solubility and mobility in the soil (Herrera et al., 2022). Large amounts of chemical $P_i$ fertilizers are applied during agricultural practices to alleviate low P availability, yet plants take up only 20–30% of the applied $P_i$ fertilizer (McDowell & Haygarth, 2024). Targeting genes that increase phosphorus use efficiency (PUE) is an alternative strategy to circumvent the long-term consequences of excessive P fertilizer in agricultural systems. Genes related to the mobilization and recycling of cellular P fractions are promising candidates for increased PUE (Han et al., 2022).

P in plants can be grouped into organic and inorganic fractions based on their chemical structure. Organic P ($P_o$) includes nucleic acids, glycerophospholipids and low-molecular-weight phospho-ester (P-ester) fractions (Suriyagoda et al., 2023; Tsujii et al., 2023). Nucleic acids represent the predominant sink (>50%) for $P_o$ in plant leaves, with approximately 50% of these present as ribosomal RNA (rRNA), followed by organellar DNA (7%), tRNA (2%) and mRNA (1%) (Busche et al., 2021). Phospholipids (PLs, P-lipids) constitute the second most abundant fraction of $P_o$ (30%) in plant cells (Busche et al., 2021). They are synthesized primarily in the endoplasmic reticulum (ER), which accounts for >60% of cellular PLs by mass (Lagace & Ridgway, 2013). Finally, low-molecular-weight P-esters comprise phosphorylated metabolites, free nucleotides and phosphorylated proteins that amount to 20% of $P_o$ in plant cells (Busche et al., 2021). The diversity of chemical structures found in low-molecular-weight P-esters makes this fraction the most diverse in plants (Busche et al., 2021).

$P_i$ is the predominant form of inorganic phosphate in plants, with a small portion existing as pyrophosphate ($P_2O_7^{4-}$) (Tsujii et al., 2023). As mentioned above, $P_i$ is neither easily accessible nor evenly distributed due to its low solubility and poor mobility in the soil (Herrera et al., 2022). $P_i$ is directly absorbed by the roots and transported within the plants through the action of membrane-localized $P_i$ transporters. Under sufficient P, up to 75% of excess cellular $P_i$ is stored in the vacuoles through the action of vacuolar transporters (Liu et al., 2015; Liu et al., 2016). Upon $P_i$ limitation, $P_i$ is exported from the vacuole to buffer changes in cytosolic $P_i$ levels (Liu et al., 2015; Liu et al., 2016; Xu et al., 2019). $P_i$ recycling, import and storage inside the vacuole are crucial for maintaining a functional level of cellular metabolism (Yoshitake & Yoshimoto, 2022).

It is crucial to control the mobilization and recycling of intracellular P, especially when external P availability fluctuates. $P_i$ mobilization and recycling strategies vary in their targets and cellular localization, as outlined in Figure 1. $P_i$ is mobilized through $P_i$ transporters on the plasma and organellar membranes. Additionally, intracellular P-containing biomolecules such as nucleic acids and PLs can be metabolized to release $P_i$ to adjust cytosolic $P_i$ concentrations (Yoshitake & Yoshimoto, 2022). Recent studies also revealed that $P_i$ can be remobilized from the cell wall (Zhu et al., 2016; Qi et al., 2022). Other aspects of P starvation responses (PSRs), such as those related to $P_i$ acquisition, transport and regulation of local and systemic P signalling, have been covered and discussed in recent reviews (Wang et al.,

2021; Yoshitake & Yoshimoto, 2022; Puga et al., 2024; Yang et al., 2024). This review will focus on intracellular $P_i$ recycling, mobilization and the corresponding regulation. Notably, many of these strategies are regulated by a central module of transcriptional activation by PHOSPHATE STARVATION RESPONSE (PHR) and suppression by (SYG1/PHO81/XPR1) SPX proteins with inositol pyrophosphates (PP-InsPs) as signals of intracellular P status (Puga et al., 2014; Wild et al., 2016; Dong et al., 2019; Zhu et al., 2019). Finally, we highlight the gaps in our current understanding of $P_i$ recycling and P sensing, and the coordination between recycling and remobilization and the potential use of the key genes from these strategies as targets for improving PUE in crops.

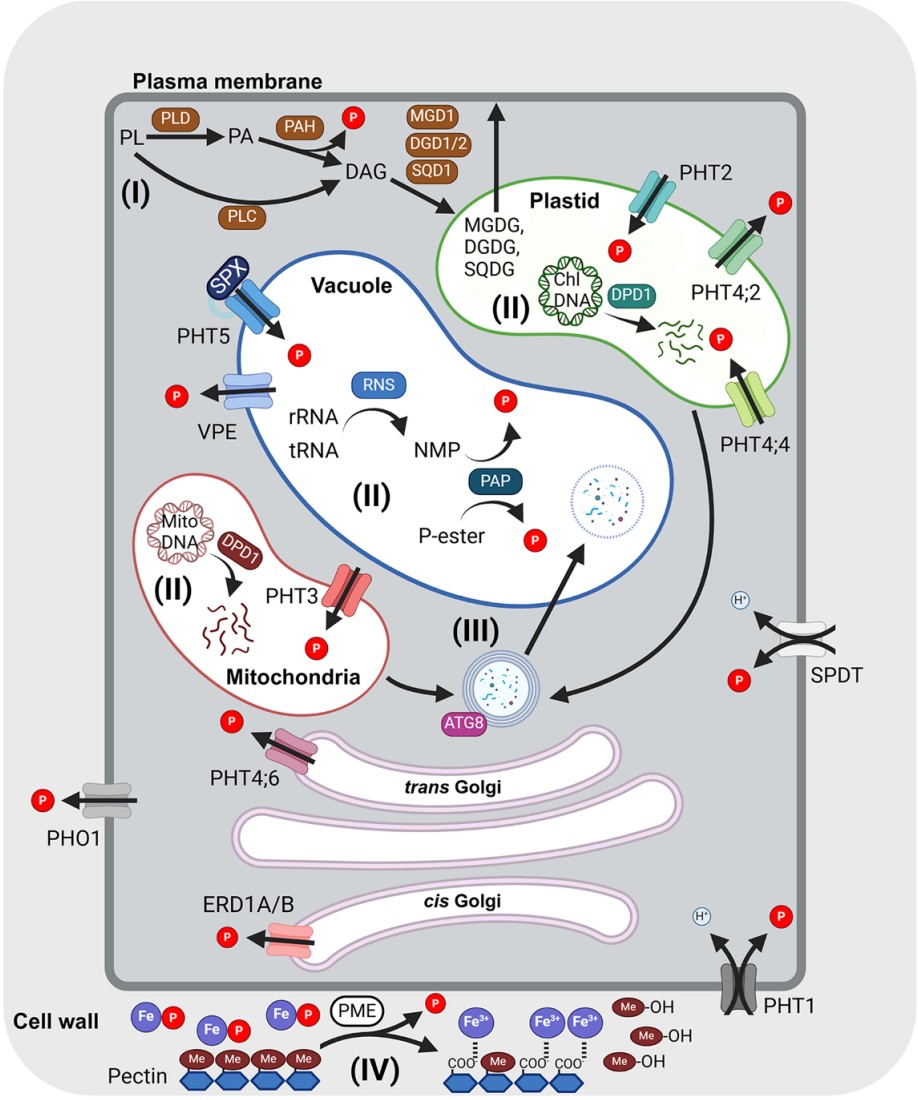

**Figure 1.** Strategies for intracellular P recycling and mobilization in plant cells.
Different pathways for intracellular $P_i$ recycling and mobilization are outlined as follows: (I) Lipid remodelling at the plasma membrane, (II) degradation of nucleic acids, (III) autophagy and (IV) $P_i$ remobilization from the cell wall. $P_i$ mobilization is mediated by PHT1 $P_i$ transporters, PHOSPHATE 1 (PHO1) and SULTR-like phosphorus distribution transporter (SPDT) across the plasma membrane, PHT2 and PHT4 in the plastids, PHT3 in the mitochondria and PHT5 and vacuolar phosphate efflux (VPE) on the vacuolar membrane. PHT4;6 and ER retention defective 1A/B (ERD1A/B) are located in the *trans*-Golgi and *cis*-Golgi, respectively. The arrows indicate the transport direction. Metabolic genes involved in P recycling are labelled as follows: autophagy-related 8 (ATG8), defective in pollen organelle DNA degradation1 (DPD1), DIGALACTOSYL DIACYLGLYCEROL DEFICIENT 1/2 (DGD1/2), pectin methyltransferase (PME), phospholipase C (PLC), phospholipase D (PLD), phosphatidic acid phosphatase (PAH), ribonuclease 2 (RNS2), sulphoquinovosyldiacylglycerol 1 (SQD1), purple acid phosphatase (PAP). Organic and inorganic phosphates are labelled as follows: diacylglycerol (DAG), digalactosyldiacylglycerol (DGDG), methanol (Me-OH), monogalactosyldiacylglycerol (MGDG), phosphatidic acid (PA), phospholipid (PL) and sulphoquinovosyldiacylglycerol (SQD) (see the text for details). This figure was created using BioRender.

## 2. $P_i$ transporters in P mobilization

$P_i$ transporters located on the plasma membrane, which carry $P_i$ in and out of cells, are primarily responsible for uptake from the soil by importing $P_i$ or exporting $P_i$ as a means to translocate $P_i$ between tissues. On the other hand, organellar $P_i$ transporters deliver $P_i$ across organellar membranes to modulate the cytosolic $P_i$ concentration and the $P_i$ concentration inside the organelles (Figure 1). The coordination of these transport activities is essential for controlling cytosolic $P_i$ concentrations. There are three types of $P_i$ transporter families located in plasma membranes: members of the $P_i$ transporter 1 (PHT1), PHOSPHATE1 (PHO1) and SULTR-like $P_i$ distribution transporter (SPDT) families (Yang et al., 2024). PHT1 members are primarily responsible for initial $P_i$ acquisition from the roots and subsequent $P_i$ allocation among various tissues and organs. PHO1 members are $P_i$ efflux transporters predominantly expressed in the root pericycle and xylem parenchyma cells for $P_i$ loading into the xylem (Hamburger et al., 2002). PHO1 members are also expressed in the seed coat, essential for transferring $P_i$ from maternal to filial tissues to sustain seed development (Vogiatzaki et al., 2017; Che et al., 2020; Ma et al., 2021; Ko et al., 2024). SPDTs are node-localized $P_i$ transporters responsible for loading $P_i$ into grains in rice (Yamaji et al., 2017) and barley (Gu et al., 2022). Knockout of rice SPDTs reduces grain $P_i$ loading and phytic acid synthesis without any penalty on the yield (Yamaji et al., 2017). Arabidopsis SPDT members are expressed in the rosette basal region and leaf petiole and preferentially allocate $P_i$ to younger leaves (Ding et al., 2020).

As to the organellar $P_i$ transporters, PHT2 transporters are localized in the chloroplasts, PHT3 transporters are in the mitochondria, PHT4 members are in the plastids or Golgi apparatus and PHT5 (or vacuolar $P_i$ transporter (VPT)) and vacuolar $P_i$ efflux (VPE) are VPTs (Yang et al., 2024). The chloroplast and mitochondrial $P_i$ transporters are essential for sustaining photosynthetic activity and ATP generation (Flugge et al., 2011; Jia et al., 2015; Raju et al., 2024). VPTs are critical in buffering cytosolic $P_i$ levels (Liu et al., 2015; Liu et al., 2016; Xu et al., 2019). In the following section, we will discuss the roles of the vacuolar and organellar $P_i$ transporters in intracellular $P_i$ remobilization and recycling.

### 2.1. Vacuolar $P_i$ transporters

Under sufficient $P_i$ supply, most intracellular $P_i$ is sequestered in the vacuoles, the largest organelle in plant cells (Yang et al., 2017). When $P_i$ supply is scarce, $P_i$ is released from the vacuoles to meet demand in the cytoplasm. Two types of VPTs mediate $P_i$ sequestration and liberation, respectively: influx transporter PHT5, responsible for $P_i$ storage inside vacuoles (Liu et al., 2015; Liu et al., 2016), and the VPE transporter, required for exporting $P_i$ from vacuoles (Xu et al., 2019). Both $P_i$ transporters belong to the major facilitator superfamily (MFS), in which PHT5 members contain an additional SPX domain at their N terminus involved in regulating their transport activity (Luan et al., 2022).

The SPX domain of the PHT5 members binds to PP-InsPs and is implicated in $P_i$ sensing and signalling. Removal of the N-terminal 229 amino acids (including the SPX domain) of PHT5 constitutively turns on its transport activity. Still, mutation of the conserved PP-InsP binding pocket in the SPX domain abolishes this activity (Luan et al., 2022). A recent study showed that loss of function of VHA-A, an essential subunit of vacuolar $H^+$-ATPase, increased the vacuolar pH value but reduced the vacuolar $P_i$ concentration (Sun et al., 2024). It is unclear how the change in the acidification of the vacuolar lumen affects the transport activity of PHT5, because its $P_i$ transport activity is independent of ATP and the $H^+$ gradient when examined in yeast vacuoles (Liu et al., 2016). One plausible explanation is that PHT5-mediated transport could be facilitated by the positive inside potential across the tonoplast. The concentration gradient would be generated by protonating the divalent $P_i$ ($HPO_4^{2-}$) to monovalent $P_i$ ($H_2PO_4^-$) inside the acidic vacuolar lumen (Massonneau et al., 2000; Versaw & Garcia, 2017). In rice, the expression of OsPHT5 (SPX-MFS1 and SPX-MSF2) was post-transcriptionally suppressed by microRNA827 (miR827) upon $P_i$ starvation (Lin et al., 2010; Wang et al., 2012). This regulation may also apply to non-Brassicales species (Lin et al., 2018). The OsPHT5 activity is shown to be modulated by its trafficking from pre-vacuolar compartments to the vacuolar membrane by interacting with the syntaxin of plants (OsSYP21 and OsSYP22) with its SPX domain (Guo et al., 2023). Unlike PHT5, the gene expression of rice VPE is upregulated by OsPHR2 under $P_i$ starvation (Xu et al., 2019).

Loss-of-function *pht5* Arabidopsis mutants led to low vacuolar $P_i$ content and necrotic leaves during P replenishment after starvation (Liu et al., 2016). Overexpression of *PHT5* resulted in over-accumulation of $P_i$ inside the vacuole, resulting in reduced cytosolic $P_i$ concentrations leading to retarded growth and upregulation of $P_i$ starvation-responsive genes even under $P_i$ sufficiency (Liu et al., 2016). Overexpression of PHT5 also retained more $P_i$ in the leaves and impaired $P_i$ allocation to flowers (Sun et al., 2023). In contrast to PHT5, overexpressing VPE in rice reduced $P_i$ accumulation in vacuoles, whereas *vpe* mutants displayed a higher vacuolar $P_i$ level (Xu et al., 2019).

### 2.2. miR399- and miR827-mediated $P_i$ transport

MicroRNA399 (miR399) and miR827 are well-studied $P_i$-starvation-induced microRNAs that regulate cytosolic $P_i$ homeostasis (Liu et al., 2014; Chien et al., 2017). *MIR399* and *MIR827* genes are evolutionarily conserved (Hsieh et al., 2009; Lin et al., 2018) and serve as long-distance signalling molecules for systemic regulation (Chien et al., 2018). MiR399 suppresses the expression of *PHO2*, which encodes a ubiquitin-conjugating E2 enzyme (Lin et al., 2008; Kuo & Chiou, 2011). PHO2 proteins localized in the ER and Golgi regulate the protein stability of PHT1 and PHO1 transporters to control $P_i$ uptake and root-to-shoot translocation activities, respectively (Liu et al., 2012; Huang et al., 2013). Overexpression of miR399 or loss of function of *PHO2* enhances $P_i$ uptake and translocation and leads to over-accumulation of $P_i$ in shoots (Aung et al., 2006; Chiou et al., 2006). MiR827 targets two different transcripts encoding SPX-domain-containing proteins, *NITROGEN LIMITATION ADAPTATION* (*NLA*) in Brassicales and *PHT5* in non-Brassicales species (Lin et al., 2018). As mentioned above, PHT5 is a vacuolar $P_i$ import transporter (Wang et al., 2012; Liu et al., 2015; Liu et al., 2016). *NLA* encodes a plasma membrane-localized ubiquitin E3 ligase belonging to the SPX-RING protein family (Lin et al., 2013). NLA regulates the degradation of PHT1 by ubiquitination-mediated endocytosis (Lin et al., 2013). Overexpression of miR827 and loss of function of *nla* mutants impaired $P_i$ remobilization from older to young leaves in rice (Wang et al., 2012) and accumulated higher amounts of $P_i$ in Arabidopsis leaves (Lin et al., 2013; Val-Torregrosa et al., 2022). Of note, the upregulation of miR399 and miR827 by low $P_i$ and the function of PHO2 and NLA in regulating $P_i$ transport are evolutionarily conserved (Lin et al., 2018).

### 2.3. PP-InsP-SPX-PHR module

PHR1 in Arabidopsis and PHR2 in rice are considered the central regulators of PSRs in plants (Rubio et al., 2001; Zhou et al., 2008). PHR1 binds to the PHR1-binding sequence (P1BS) cis-element, preferentially found in genes responding to $P_i$ starvation. The PHR1 transcript and protein level are weakly responsive to $P_i$ starvation. However, PHR1-mediated upregulation of PSR is repressed through its interaction with SPX proteins (Puga et al., 2014; Wang et al., 2014b). Interestingly, several SPX transcripts are upregulated by PHR during $P_i$ starvation, which indicates that SPX proteins are involved in a negative feedback regulatory loop with PHR (Puga et al., 2014).

Recent studies have identified PP-InsPs as signalling molecules for sensing intracellular P status (Wild et al., 2016; Dong et al., 2019; Zhu et al., 2019). PP-InsPs were able to bind to the SPX-containing proteins from various organisms (Wild et al., 2016). The genetic analyses of genes encoding diphosphoinositol pentakisphosphate kinases VIH1/2 revealed that bis-diphosphoinositol tetrakisphosphate (1,5-InsP8) acts as an intracellular signalling molecule that translates the cellular $P_i$ status to PSR in plants (Dong et al., 2019; Ried et al., 2021). Under sufficient P, the binding of InsP8 to SPX proteins promotes its interaction with PHR1 to prevent its transcriptional activation of PSR genes. Conversely, PHR1 dissociates from SPX1 when the InsP8 level drops under P starvation, which allows PHR1 to bind to the P1BS sites to activate PSR genes.

### 2.4. Other organelle $P_i$ transporters

Chloroplasts and mitochondria carry out vital metabolic reactions, including photosynthesis, carbon assimilation, respiration and oxidative phosphorylation (Flugge et al., 2011), which are regulated by optimal $P_i$ concentrations. $P_i$ is delivered into chloroplasts and mitochondria by PHT2, PHT3 and PHT4 transporters (Versaw & Garcia, 2017). These organellar $P_i$ transporters mediate the distribution of $P_i$ to balance its concentration between the cytosol and organelles. In Arabidopsis, AtPHT2;1 is a low-affinity $P_i$ transporter located in the chloroplast inner envelope membrane whose expression is independent of external $P_i$ supply but induced by light (Versaw & Harrison, 2002). Characterization of the loss-of-function atpht2;1 mutant revealed that PHT2;1 contributes to $P_i$ import into chloroplasts and eventually affects the accumulation of $P_i$ in leaves and the allocation of $P_i$ throughout the plant (Versaw & Harrison, 2002; Raju et al., 2024). Similar results were observed for rice OsPHT2;1 (Liu et al., 2020).

Arabidopsis has six PHT4 members. Except for PHT4;6, they are localized in the photosynthetic and/or heterotrophic plastids (Guo et al., 2008), among which PHT4;2 has a physiological role in $P_i$ export from root plastids (Irigoyen et al., 2011). Although all the PHT4s mediate $P_i$ transport in yeast cells (Guo et al., 2008), interestingly, AtPHT4;4 exhibited ascorbate uptake activity (Miyaji et al., 2015). PHT4;6 and ER Defective 1A (ERD1A) and ERD1B reside in the Golgi apparatus and are involved in $P_i$ release from the trans- and cis-Golgi compartment, respectively (Cubero et al., 2009). Loss of function of PHT4;6 reduced cytosolic $P_i$ content but enhanced $P_i$ reallocation to the vacuole and activated disease resistance mechanisms (Hassler et al., 2012). In contrast, the erd1a mutant altered cell wall monosaccharide composition with increased apoplastic $P_i$ export activity, likely due to exocytosis (Hsieh et al., 2023). PHT4;6 is also required for ammonium and sugar metabolism and mediates dark-induced senescence (Hassler et al., 2016).

The PHT3 transporters in the inner mitochondrial membrane operate $P_i$ translocation into the mitochondrial matrix (Nakamori et al., 2002; Hamel et al., 2004). Overexpression of AtPHT3;1 accumulated higher ATP content, faster respiration rate and more reactive oxygen species than wild-type plants, severely hampering plant development (Jia et al., 2015). The expression of Arabidopsis PHT3 transporters was upregulated by salinity, but overexpressing PHT3 displayed increased sensitivity to salt stress, likely due to the disturbance of ATP and gibberellin metabolism (Zhu et al., 2012).

## 3. Intracellular P recycling

Besides increasing external $P_i$ uptake and release from the vacuole to overcome $P_i$ starvation, $P_i$ recycling is an additional vital system that salvages $P_i$ from many intracellular components that contain P, including from degradation of nucleic acids, membrane lipid remodelling, P remobilization from the cell wall and organelle degradation via catabolic enzymes (labelled I–IV in Figure 1).

### 3.1. Phosphate scavenging from nucleic acids

Scavenging the $P_i$ from the nucleic acid in leaves during $P_i$ starvation involves the action of hydrolytic enzyme nucleases (RNases) and purple acid phosphatases (PAPs) (Bassham & MacIntosh, 2017; Yoshitake et al., 2022). rRNA is the predominant form of RNA found in most cells; it makes up about 80% of cellular RNA (Palazzo & Lee, 2015). The RNS2, a subclass of RNase T2 localized in the vacuoles and ER (Floyd et al., 2016), converts RNA into nucleotide monophosphates, which are then dephosphorylated by PAPs. In rice, the expression of both RNSs and PAPs is induced by $P_i$ starvation, which hydrolyses 60–80% of the total RNA in flag leaves to release and remobilize $P_i$ to developing grains (Jeong et al., 2017; Gho et al., 2020). Other than rRNA, specific transfer RNA (tRNA)-derived fragments (tRFs) from the tRNA cleavage (i.e., tRNA[Gly] and tRNA[Asp]) by AtRNSs (RNS1–RNS3) were accumulated under $P_i$ starvation (Hsieh et al., 2009; Megel et al., 2019). In addition to a housekeeping role, RNase-mediated RNA degradation participates in $P_i$ recycling during $P_i$ starvation.

Organelle DNA (orgDNA), which encodes a small genome with multiple copies in vegetative tissues, could also be a source of $P_i$ when degraded (Sakamoto & Takami, 2024). Arabidopsis organellar exonuclease, defective in pollen orgDNA degradation 1 (AtDPD1), operates plastid and mitochondrial DNA degradation during leaf senescence and pollen development (Takami et al., 2018). Loss of function of AtDPD1 inhibits orgDNA degradation under $P_i$ starvation, which maintains a high copy number of chloroplast DNA, leading to compromised PSR gene expression and P remobilization from old to young leaves (Takami et al., 2018; Islam et al., 2024).

PAPs are $P_i$ starvation-induced acid phosphatases, which hydrolyse phosphomonoesters from various organic P compounds to release $P_i$ at acidic pHs (Robinson et al., 2012). PAPs are localized in intracellular compartments or secreted to extracellular spaces. The secreted PAPs are associated with the root surface and aid in $P_i$ solubilization in the rhizosphere (Wang et al., 2014a; O'Gallagher et al., 2022). Overexpression of PAP genes improves plant biomass and total P accumulation when $P_o$ (e.g., ATP, DNA) is supplied as the sole external P source (Deng et al., 2020). Besides conventional phosphatase activity, some PAPs also display phosphodiesterase (Olczak et al., 2000; Wang et al., 2014a) or phytase activity (Bhadouria et al., 2017; Kong et al., 2018). A broad substrate

specificity and widespread localization profiles of PAPs may help plants maintain intracellular $P_i$ balance.

### 3.2. Phosphate scavenging from membrane lipid remodelling

Membrane lipid remodelling is one of the most dramatic metabolic responses to $P_i$ starvation. It replaces the PLs, such as phosphatidyl-cholines, phosphatidylglycerol and phosphatidylethanolamine (PE), with galactolipid digalactosyldiacylglycerol (DGDG) and sulphoquinovosyldiacylglycerol (SQDG) to release $P_i$ with minimal or no damage to membrane function (Lambers et al., 2012). Phospholipase C (PLC), phospholipase D (PLD) and phospha-tidic acid phosphatase homolog (PAH) are the major enzymes contributing to PL hydrolysis (Nakamura, 2013). PLDs work by hydrolysing the phosphodiester bond of PLs to produce phosphatidic acid (PA) and polar head groups (Li et al., 2006b). PAH then dephosphorylates PAs to form diacylglycerol (DAG) and releases $P_i$ (Nakamura et al., 2009). PLCs behave differently from PLDs as they produce DAG in a single step to release the P-containing polar head group (Nakamura et al., 2005; Gaude et al., 2008). In Arabidopsis, two NON-SPECIFIC PLCs (NPC4, 5) and PLDζ (PLDζ1 , PLDζ2) are endomembrane localized and their expression is highly induced by $P_i$ starvation (Li et al., 2006c; Li et al., 2006a; Gaude et al., 2008). Impairment of both *PLDζ2* and *NPC4* (*npc4pldζ2*), which increases PE but decreases DGDG, impedes primary root growth and root hair density under $P_i$ deprivation (Su et al., 2018). Mutation in the Arabidopsis PAH, as seen in *pah1/pah2* double mutant, suppressed membrane lipid remodelling and showed root growth defects as seen in *npc4pldζ2*, indicating PL hydrolysis enzymes are important in the $P_i$ recycling under Pi starvation (Nakamura et al., 2009).

Synthesis of non-P-containing galactolipids and sulpholipids using DAG is another alternative step in membrane lipid remod-elling during $P_i$ starvation (Nakamura, 2013). SQDG biosynthe-sis is mediated by uridine diphosphate (UDP)-sulphoquinovose synthase 1 and 2 (SQD1 and 2). SQD1 catalyses the assembly of UDP-sulphoquinovose (SQ) via UDP-glucose and sulphite (Sanda et al., 2001), and then SQD2 functions in transferring the sulpho-quinovose of UDP-SQ to DAG to generate SQDG (Yu et al., 2002). The expression of both SQD1 and 2 is upregulated by $P_i$ limitation (Yu et al., 2002; Jeong et al., 2017). Knockout of AtSQD2 decreases the amount of SQDG and reduces fresh weight under $P_i$ starvation (Okazaki et al., 2013).

Monogalactosyldiacylglycerol (MGDG) is synthesized from DAG by MGDG synthase; subsequently, DGDG can be further synthesized from MGDG by DGDG synthase (Nakamura, 2013). Arabidopsis has two types of MGDG synthase, Type A (MGD1) and Type B (MGD2 and MGD3) (Awai et al., 2001). MGD1 is expressed in green tissues and localized in the inner envelope of chloroplasts and plays pivotal roles in photosynthetic membrane biogenesis (Jarvis et al., 2000; Kobayashi et al., 2007). In contrast, MGD2 and MGD3 localize on the outer envelope membranes of plastids, and their expressions are strongly activated by $P_i$ starvation (Awai et al., 2001; Jeong et al., 2017). Arabidopsis has two DGDG synthases, AtDGD1 and AtDGD2, and both are induced by $P_i$ deficiency (Kelly & Dormann, 2002). In the *dgd1* mutant, the DGDG level is significantly reduced, and its growth is impaired under P-deficient conditions (Hartel et al., 2000).

A large number of genes involved in lipid remodelling contain the P1BS motifs in their promoter region, for example, NCP4/5, PLDζ2, PAH1/2, MGD2/3 and SQD1/2 (Pant et al., 2015). Loss-of-function Arabidopsis *phr1* mutants showed reduced expression of these genes and changes in lipid composition in response to P deficiency (Pant et al., 2015), reinforcing the role of PHR1 in membrane $P_i$ recycling.

### 3.3. Demethylation of pectin enhances cell wall P remobilization

In addition to the intracellular P, pectin in the cell wall has been proposed to contribute to P remobilization from cell wall under $P_i$ starvation (Zhu et al., 2015; Qi et al., 2022). The quasimodo1 (*qua1*) mutant encoding a glycosyltransferase for pectic synthesis has low pectin content and is more sensitive to P deficiency than the wild-type control (Zhu et al., 2015). The carboxyl groups in homogalacturonan (HG), the most abundant pectin subtype, can be demethylated by pectin methylesterase (PME), which liberates protons and methanol and produces a carboxylate group (Wormit & Usadel, 2018). It was hypothesized that the negatively charged carboxylate groups on the HG in pectin have a high affinity for $Al^{3+}$ and $Fe^{3+}$, which may potentially solubilize P sequestered as the forms of $AlPO_4$ and $FePO_4$ within the cell wall (Zhu et al., 2015). OsPME14, the only member of rice PMEs induced by P starvation, may facilitate root cell wall $P_i$ remobilization (Qi et al., 2022). Overexpressing OsPME14 showed higher PME activity with more cell wall Fe accumulation and soluble P in the root compared to the wild type (Qi et al., 2022). PME activity is regulated by several factors, such as nitric oxide (Zhu et al., 2017), ethylene (Zhu et al., 2016; Zhang et al., 2021) and abscisic acid (Zhu et al., 2018). Nevertheless, the direct evidence for PME-mediated cell wall P remobilization is still lacking.

### 3.4. $P_i$ scavenging by autophagy

Autophagy is an intracellular degradation process in vacuoles for bulk protein and organelles to recycle nutrients under starvation (Nakatogawa, 2020). The proteins encoded by *autophagy-related genes* (*ATG*) participate in autophagosome induction, membrane delivery, vesicle nucleation, cargo recognition and phagophore expansion and closure (Nakatogawa, 2020). Most *ATG* genes in Arabidopsis are highly induced by nitrogen starvation but are moderately upregulated by $P_i$ starvation (Chiu et al., 2023). Only *AtATG8f* and *AtATG8h* were upregulated in the $P_i$-deprived root, which is mediated by AtPHR1 indirectly (Lin et al., 2023), suggesting second-wave transcriptional regulation. Low $P_i$ promotes the autophagic flux preferentially in the differential zone of the Arabidopsis root (Naumann et al., 2019; Chiu et al., 2023). Mutation of ATG genes (*ATG5*, *7*, *9* and *10*) reduced the $P_i$ translocation to retain more $P_i$ in the root and inhibited meristem development under $P_i$ sufficiency. Autophagy-deficient mutants, *atg2* and *atg5*, showed early depleted $P_i$ and severe leaf growth defects under $P_i$ starvation (Yoshitake & Yoshimoto, 2022).

ER stress-induced ER-phagy was observed during the early phase of $P_i$ starvation, contributing to $P_i$ recycling and suppressing membrane lipid remodelling, a late PSR (Yoshitake & Yoshimoto, 2022). In the $P_i$-starved root apex, ER stress-induced ER-phagy was also observed; however, it is regarded as a sign of local $P_i$ sensing rather than a means for $P_i$ recycling (Naumann et al., 2019). Furthermore, rubisco-containing body-mediated chlorophagy, which contains chloroplast stroma, was formed when $P_i$ limitation was coupled with N and C availability (Yoshitake et al., 2021). Autophagy of organelles plays a role in multiple nutrients recycling, although the mechanism of $P_i$ recycling is relatively unclear.

## 4. Perspectives and challenges

In plants, P$_i$ recycling involves complex metabolic cascades regulated in response to cellular P-level changes. Studies on P$_i$ recycling have revealed the identity of numerous enzymes that are utilized to convert the diverse biomolecules that comprise cellular P$_o$ to P$_i$. Several questions remain: (1) Is there a preference for which P$_o$ fraction to recycle under a limited P supply? However, P$_o$, such as rRNA and PLs, are not completely depleted under limited P supply but are instead regulated due to their essential cellular function. Whether there is a preference regarding which P$_o$ fraction recycles P$_i$ under a limited P supply has yet to be explored. With regard to the presentation of P distribution in the current literature, it is worth noting that the most frequently cited studies that provide detailed measurements of P$_i$ content and different types of P$_o$ from plants were conducted decades ago. Re-analysing P fractions from plant tissues, cells and organelles through the lens of current advanced techniques, such as mass spectrometry, biosensors and imaging techniques, with spatial and temporal resolution, will provide an up-to-date reference for investigating the effects of P$_i$ recycling on P distribution.

Although we have discussed P$_i$ recycling and mobilization as separate strategies that allow plants to supply and deliver P$_i$, both methods must act in concert to maintain whole-plant P homeostasis in response to changing P availability. However, a detailed account that describes a coordinated function between recycling and mobilization in response to P$_i$ availability has yet to be formulated. Furthermore, both P$_i$ recycling and P$_i$ mobilization operate in a complex network that must be coordinated to balance the internal cellular P$_i$ level with the external P$_i$ supply. Several components of P recycling and mobilization are known to be controlled by the PP-InsP-SPX-PHR1 module, which suggests a common mode of regulation. The presence of PHR-independent regulation of P recycling and mobilization suggests additional mechanisms remain to be uncovered. Resolving the gaps in our knowledge regarding P recycling, mobilization and signalling will offer information that may be invaluable for ecological and agricultural applications. The genes involved may be used as candidate targets for gene editing or as breeding markers for the future improvement of crop PUE to achieve sustainable agriculture. Nevertheless, as implied by the known interaction between nutrients, such as phosphorus, iron, zinc and nitrogen, balance with other nutrients should be considered when enhancing crop PUE. In the long run, the extent of the potential impacts of high PUE crops on ecology, such as soil microbial, faunal or even other plant communities, should also be evaluated.

**Financial support.** The authors would like to thank Academia Sinica (AS-GCS-112-L03 and AS-IA-113-L06) and the National Science and Technology Council (NSTC 112-2311-B-001-040-MY3), Taiwan for funding to T.-J. C.'s laboratory.

**Competing interest.** The authors declare none.

**Author contributions.** Chih-Pin Chiang and Joseph Yayen these authors contributed equally. C.-P. C., J. Y. and T.-J. C. conceptualized the review outline. C.-P. C. and J. Y. prepared the figures and wrote the original draft. T.-J. C. compiled, reviewed and edited the manuscript. All authors contributed to the critical revision and its final approval.

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
