## [Reviewer Report]

This is a well-written review of the transport and metabolic recycling processes that plants utilize to respond to phosphorus starvation. I appreciate the timely inclusion of remobilization of Pi from cell wall pectin and the role of autophagy for Pi scavenging.

The only item for the authors to consider is the statement on page 7, line 4: “It is unclear how the change in acidification of the vacuolar lumen affects the transport activity of PHT5 because its transport activity is independent of ATP and the H+ gradient when examined in yeast vacuoles.” One plausible explanation is that transport is facilitated by the positive-inside potential across the tonoplast. Moreover, if there is specificity for the divalent form of Pi then it would be protonated in the acidic lumen and thereby alter the effective concentration gradient. This concept was described previously (Planta 2000, 211:390-395, and Current Opinion in Plant Biology 2017, 39:25–30) as a means to concentrate Pi in the lumen. The net effect on transport may not be readily detected in yeast vacuoles under typical assay conditions.

---

## [Reviewer Report]

This short review paper focuses on intracellular P recycling and mobilization in response to P deficiency in plants. Each part is concisely summarized based on previous publications. I only have a few comments as listed below.

1. On page 5, the authors mentioned SPDT in rice and barley, but the reference on barley was not cited. Furthermore, the reference on rice SPDT should be 2017, but not 2016. The controversy results on rice SPDT should be discussed a little more based on recent publication

2. The introduction part could be slightly reduced, because similar thing is described in each section.

3. I suggest to remove Pi remobilization from cell wall pectin part on page 15 because no convincing evidence supports this issue.

---

## [Editor Report]

Dear authors,

The manuscript was reviewed by two independent reviewers. Both are very positive, but make some suggestions. It is advisable to try to take these points into account in a minor revision. Thank you very much for your contribution to the Research Topic “Quantitative approaches to cellular aspects of plant ion homeostasis”.

Best regards, Ingo Dreyer

---

## [Editor Report]

Dear authors,

thank you for the careful revision of the manuscript. And thanks again for your valuable contribution to the Research Topic “Quantitative approaches to cellular aspects of plant ion homeostasis”. It is highly appreciated.

Best regards, Ingo